Th2 cytokine bias induced by silver nanoparticles in peripheral blood mononuclear cells of common bottlenose dolphins (Tursiops truncatus)

Li Wen-Ta 1 2
Wang Lei-Ya 1 2
Chang Hui-Wen 1 2
Yang Wei-Cheng 2
Lo Chieh 3
Pang Victor Fei 1 2
Chen Meng-Hsien 4
Jeng Chian-Ren crjeng@ntu.edu.tw 1 2
1 Graduate Institute of Molecular and Comparative Pathobiology, National Taiwan University , Taipei , Taiwan
2 School of Veterinary Medicine, National Taiwan University , Taipei , Taiwan
3 Farglory Ocean Park , Hualien , Taiwan
4 Department of Oceanography and Asia-Pacific Ocean Research Center, National Sun Yat-Sen University , Kaohsiung , Taiwan
Brander Susanne
Electronic publication date: 2018 Sep 17
Publication date: 2018
Volume: 6
Electronic Location ID: e5432
Received 2018 Feb 14; Accepted 2018 Jul 20
Copyright: ©2018 Li et al.
Copyright year: 2018
Copyright holder: Li et al.
License: This is an open access article distributed under the terms of the Creative Commons Attribution License, which permits unrestricted use, distribution, reproduction and adaptation in any medium and for any purpose provided that it is properly attributed. For attribution, the original author(s), title, publication source (PeerJ) and either DOI or URL of the article must be cited.
License URL: https://creativecommons.org/licenses/by/4.0/

Keywords: Cetacean, Silver nanoparticles (AgNPs), Immunotoxicity, Cytokine, Th2 bias, qRT-PCR

Funding: Ministry of Science & Technology MOST 106-2313-B-002-054 The study was supported by the Ministry of Science & Technology, Taiwan under Grant MOST 106-2313-B-002-054. The funders had no role in study design, data collection and analysis, decision to publish, or preparation of the manuscript.

==============================
Background

Silver nanoparticles (AgNPs) have been widely used in many commercial products due to their excellent antibacterial ability. The AgNPs are released into the environment, gradually accumulate in the ocean, and may affect animals at high trophic levels, such as cetaceans and humans, via the food chain. Hence, the negative health impacts caused by AgNPs in cetaceans are of concern. Cytokines play a major role in the modulation of immune system and can be classified into two types: Th1 and Th2. Th1/Th2 balance can be evaluated by the ratios of their polarizing cytokines (i.e., interferon [IFN]-γ/Interleukin [IL]-4), and animals with imbalanced Th1/Th2 response may become more susceptible to certain kinds of infection. Therefore, the present study evaluated the in vitro cytokine responses of cetacean peripheral blood mononuclear cells (cPBMCs) to 20 nm citrate-AgNPs (C-AgNP20) by quantitative reverse transcriptase polymerase chain reaction (qRT-PCR).

Methods

Blood samples were collected from six captive common bottlenose dolphins (Tursiops truncatus). The cPBMCs were isolated and utilized for evaluating the in vitro cytokine responses. The cytokines evaluated included IL-2, IL-4, IL-10, IL-12, interferon (IFN)-γ, and tumor necrosis factor (TNF)-α. The geometric means of two housekeeping genes (HKGs), glyceraldehyde 3-phosphate dehydrogenase (GAPDH) and β2-microglobulin (B2M), of each sample were determined and used to normalize the mRNA expression levels of target genes.

Results

The ratio of late apoptotic/necrotic cells of cPBMCs significantly increased with or without concanavalin A (ConA) stimulation after 24 h of 10 µg/ml C-AgNP20 treatment. At 4 h of culture, the mRNA expression level of IL-10 was significantly decreased with 1 µg/ml C-AgNP20 treatment. At 24 h of culture with 1 µg/ml C-AgNP20, the mRNA expression levels of all cytokines were significantly decreased, with the exceptions of IL-4 and IL-10. The IFN-γ/IL-4 ratio was significantly decreased at 24 h of culture with 1 µg/ml C-AgNP20 treatment, and the IL-12/IL-4 ratio was significantly decreased at 4 or 24 h of culture with 0.1 or 1 µg/ml C-AgNP20 treatment, respectively. Furthermore, the mRNA expression level of TNF-α was significantly decreased by 1 µg/ml C-AgNP20 after 24 h of culture.

Discussion

The present study demonstrated that the sublethal dose of C-AgNP20 (≤1 µg/ml) had an inhibitory effect on the cytokine mRNA expression levels of cPBMCs with the evidence of Th2 cytokine bias and significantly decreased the mRNA expression level of TNF-α. Th2 cytokine bias is associated with enhanced immunity against parasites but decreased immunity to intracellular microorganisms. TNF-α is a contributing factor for the inflammatory response against the infection of intracellular pathogens. In summary, our data indicate that C-AgNP20 suppresses the cellular immune response and thereby increases the susceptibility of cetaceans to infection by intracellular microorganisms.

Introduction

The application of silver nanoparticles (AgNPs) in industry and in consumer products has increased, and the production of AgNPs and the number of AgNP-containing products will increase over time (Massarsky, Trudeau & Moon, 2014). AgNPs can be released during the production, transport, decay, use, and/or disposal of AgNP-containing products, subsequently draining into the surface water and then accumulating in the marine environment (Farre et al., 2009; Walters, Pool & Somerset, 2014). Therefore, the increasing use and growing production of AgNPs, as potential sources of Ag contamination, raise public concerns about the environmental toxicity of Ag (Li et al., 2018a). Previous research has demonstrated that AgNPs can precipitate in marine sediments, be ingested by benthic organisms (such as benthic invertebrate species), enter and be transferred from one trophic level to the next via the food chain, and thereby cause negative effects on the animals at different trophic levels, such as algae, invertebrates and fishes (Buffet et al., 2014; Farre et al., 2009; Gambardella et al., 2015; Huang, Cheng & Yi, 2016b; Wang et al., 2014). Previous studies have demonstrated that AgNPs are toxic to all tested marine organisms in a dose-dependent manner, indicating that AgNPs may have negative effects on marine organisms at different trophic levels of the marine environment. Immunotoxic effects of AgNPs have been demonstrated in some aquatic animals such as Nile tilapia and mussel (Gagne et al., 2013; Thummabancha, Onparn & Srisapoome, 2016). Nevertheless, to date the potential toxicity of AgNPs on marine mammals such as cetaceans has not been sufficiently studied.

AgNPs have been demonstrated to cause several negative effects, such as hepatitis, nephritis, neuron cell apoptosis, and alteration of gene expression of the brain, on laboratory mammals (Espinosa-Cristobal et al., 2013; Sardari et al., 2012; Shahare & Yashpal, 2013). In vitro studies using different cell lines have also indicated that AgNPs can cause damage to DNA, cell membranes, and mitochondria through reactive oxygen species (ROS) dependent/independent pathways and further induce cytotoxicity and genotoxicity (Kim & Ryu, 2013). In addition, previous studies conducted in laboratory mammals, including mice and rats, have demonstrated that AgNPs can enter the blood circulation through alimentary tracts and then deposit in multiple organs (Espinosa-Cristobal et al., 2013; Lee et al., 2013; Shahare & Yashpal, 2013; Van der Zande et al., 2012). Considering the negative effects of AgNPs and the presence of AgNPs in blood circulation, the negative effects of AgNPs on leukocytes should be of concern. Previous studies have demonstrated that AgNPs can cause several negative effects on human polymorphonuclear leukocytes (PMNs) and peripheral blood mononuclear cells (PBMCs). These studies demonstrated that AgNPs can cause morphological alterations, cytotoxicity, atypical cell death, inhibition of de novo protein synthesis, increased production of the CXCL8 chemokine (IL-8), and impaired lysosomal activity in human neutrophils (Poirier et al., 2014; Poirier, Simard & Girard, 2016; Soares et al., 2016). Only a few studies have investigated the toxicity of AgNPs on human PBMCs, which have shown that AgNPs can cause cytotoxicity and functional perturbations, including inhibition of proliferative activity and cytokine production (Franco-Molina et al., 2016; Ghosh et al., 2012; Huang et al., 2016a; Orta-Garcia et al., 2015; Paino & Zucolotto, 2015; Shin et al., 2007).

The environmental contamination level of AgNPs is expected to increase greatly in the near future, and cetaceans, as the top predators in the ocean, will suffer the potentially negative impacts caused by AgNPs. Besides, immunotoxic effects of AgNPs have been demonstrated previously in humans and aquatic animals. Therefore, it is crucial to investigate the immunotoxic effects caused by AgNPs in cetaceans. Generally, in vivo experiments are rarely feasible, and the ethical issues concerning the study of immunotoxic effects caused by environmental contaminants in cetaceans are difficult to overcome, so in vitro study using blood samples from captive cetaceans would be a logical and crucial approach (Beineke et al., 2010; Desforges et al., 2016). Cytokines play a major role in the modulation of the immune system, including lymphocyte proliferation/differentiation, lymphoid development, cell trafficking, and inflammatory response through the interactions between the cytokines themselves and the surface receptors of many different cells (Owen et al., 2013; Tizard, 2013a). Previous studies have found that the sequence homology of cytokines among terrestrial and aquatic mammals is low, but conserved molecule regions can still be found on biologically active areas in marine mammals, such as the receptor binding sites of cytokines, suggesting a conserved biological activity of cytokines in both terrestrial and aquatic mammal species (Beineke et al., 2004; Beineke et al., 2010). Therefore, the functions of cytokines on the immune system in cetaceans may be similar to those in mice and humans. Cytokines can be classified into two groups, Th1 and Th2, and their secretion pattern is associated with the balance of Th1 and Th2 responses (Kidd, 2003; Owen et al., 2013; Tizard, 2013b). Th1 response promotes cell-mediated immune response and thus enhances the immunity against intracellular microorganisms, such as Toxoplasma gondii and Brucella spp., and a variety of viruses. In contrast, Th2 response is associated with enhanced immunity against parasites but decreased immunity to intracellular microorganisms (Owen et al., 2013; Tizard, 2013b). The Th1/Th2 balance can be evaluated by the ratios of their polarizing cytokines (i.e., interferon [IFN]-γ/interleukin [IL]-4), and animals with imbalanced Th1/Th2 response (Th1/Th2 polarization) may become more susceptible to certain kinds of infection (Owen et al., 2013; Raphael et al., 2015; Tizard, 2013b).

Cytokine profiling is still a relatively new field of immunotoxicology in cetaceans, and thus the enzyme-linked immunosorbent assay (ELISA) kit is not widely used for cytokine profiling in cetaceans (Desforges et al., 2016). Hence, it is more feasible to study the cytokine profiling by molecular biology (i.e., quantitative reverse transcriptase polymerase chain reaction; qRT-PCR). Therefore, the present study evaluates the in vitro cytokine responses of cPBMCs to C-AgNP20 by qRT-PCR. The cytokines measured were as follows: polarizing cytokines of Th1 (IL-12 and IFN-γ) and Th2 (IL-4), and some pro- and anti-inflammatory cytokines (IL-2, IL-10, and tumor necrosis factor [TNF]-α).

Material and Methods

AgNPs characterization

Considering the extensive use of 20 nm citrate-AgNPs (C-AgNP20) in recently reported studies of cetacean and human blood cells (Huang et al., 2016a; Li et al., 2018b; Poirier et al., 2014; Poirier, Simard & Girard, 2016), commercial C-AgNP20 (Pelco® citrate Biopure™ silver; Ted Pella, Redding, CA, USA) was chosen. The C-AgNP20 had been extensively washed (without centrifugation) so that the level of trace elements becomes less than 0.000001%. Transmission electron microscopy (TEM) for determining surface area and size/shape distributions, UV-visible spectroscopy for measuring the optical properties, particle hydrodynamic diameter and zeta potential, and dynamic light scattering (DLS) for determining the size distribution were performed according to the manufacturer’s instructions and previous studies (Poirier et al., 2014; Poirier, Simard & Girard, 2016). The endotoxin level of C-AgNP20 suspension was examined by ToxinSensor™ Single Test Kit (GenScript, Piscataway, NJ, USA), and it was lower than or equal to 0.015 EU/ml. For characterization, the C-AgNP20 obtained from the manufacturer were suspended in complete RPMI-1640 medium (RPMI-1640 (Gibco, New York, NY, USA) with 10% fetal bovine serum, 2mM L-glutamine, 50 IU penicillin, and 50 µg streptomycin) at a concentration of 50 µg/ml, and then examined using a JEM-1400 (JEOL, Tokyo, Japan) TEM. The size distribution and zeta potential of the C-AgNP20 were determined through Zetasizer Nano-ZS (Malvern Instruments Inc., Westborough, MA, USA) (Table 1). Measurements were performed by using 100 and 500 µg/ml C-AgNP20 in two mM citrate buffer (pH 7.4). The C-AgNP20 were diluted to one, 10, and 100 µg/ml with two mM citrate buffer and instantly used for subsequent experiments. Two mM citrate buffer was used as a vehicle control (0 µg/ml C-AgNP20).

Table 1 The size distribution and zeta potential of the 20 nm citrate-AgNPs (C-AgNP20).

Concentration (µg/ml)	100	500	
Z-Average (nm)	26.62 ± 0.15	26.54 ± 0.08	
Size (nm) (intensity)	30.27 ± 0.18 (100%)	29.64 ± 0.30 (100%)	
Zeta potential (mV)	−38.97 ± 1.33	−44.2 ± 1.35	
PdI	0.12 ± 0.00	0.11 ± 0.01	
Notes.

Results shown are means ± SD from three different lectures.

NP were suspended in 2 nM citrate buffer and measurements performed at room temperature.

PdI, Poly-dispersity Index.

Animals and blood sample collection

All procedures involving animals were conducted in accordance with international guidelines, and the protocol has been reviewed and approved by the Council of Agriculture of Taiwan (Approval number 1051700175). Voluntary blood samples were obtained from six clinically healthy bottlenose dolphins (Tursiops truncatus) with confirmation by physical examination, complete blood count, and biochemistry on a monthly basis from 2015 to 2017 at Farglory Ocean Park. Forty millilitres of heparin-anticoagulated whole blood were collected, stored, and shipped at four °C within 8 h for subsequent experiments.

Isolation of cPBMCs

Cetacean peripheral blood leukocytes (cPBLs) were collected by slow spin method with minor modifications (Bossart et al., 2008). The isolated cPBLs were resuspended in RPMI-1640 (Gibco, New York, NY, USA) with 10% ethylenediaminetetraacetic acid (EDTA) and subsequently used in the isolation of cPBMCs by density gradient centrifugation method. After centrifugation at 1, 200 × g for 30 min at 20 °C, the cPBMCs were collected from the cell layer between the RPMI-1640 (Gibco) and Ficoll-Paque PLUS (GE Healthcare, Uppsala, Sweden), washed with RPMI-1640 twice, resuspended to a final concentration of 1 × 107 cells/ml in complete RPMI-1640 media, and kept on ice until they were utilized in subsequent experiments. The cell viability of cPBMCs was determined by the trypan blue exclusion method using a hemacytometer, and the cell purity based on the cell size (forward-scattered light; FSC) and inner complexity (side-scattered light; SSC) of cPBMCs were determined by FACScalibur flow cytometry (BD, CA, USA). The cPBMCs with higher than 90% viability and 80% purity were used in this study.

Determination of the sub-lethal dose of C-AgNP20 on cPBMCs with/without concanavalin A (ConA)

The cytotoxicity of C-AgNP20 on cPBMCs was evaluated by the Annexin V-FITC/PI Apoptosis Detection Kit (Strong Biotech, Taipei, Taiwan) according to the manufacturer’s instructions. Freshly-isolated cPBMCs were seeded in 96-well plates at a density of 5 × 105 cells/well and exposed to C-AgNP20 at concentrations of 0, 0.1, 1.0 and 10 µg/ml with or without two µg/ml ConA (Sigma-Alderich, St. Louis, MO, USA). After 24 h of culture, cells were collected and resuspended in binding buffer for further analysis by FACScalibur flow cytometry (BD). The percentages of early apoptotic (PI—and Annexin +) and late apoptotic/necrotic cells (PI + and Annexin +) were determined. A total of 8,000 events/sample were acquired. The sub-lethal doses of C-AgNP20 for cPBMCs were determined and subsequently used in the cytokine expression assay.

qRT-PCR efficiency of each primer sets

The primer sets used in cytokine expression assay are summarized in Table 2. The amplification efficiency (E) of qRT-PCR with each primer set was evaluated by the slope and R2 of standard curves using the equation: E = 10−(1∕slope) − 1 (Svec et al., 2015). The standard templates for qRT-PCR with target primer sets were prepared by serial dilution of PCR products, which were amplified from cDNA samples of isolated cPBMCs with target primer sets. The PCR product was 500-fold diluted with subsequent six steps of serial 10-fold dilutions, and subsequently used for qRT-PCR. The Cycle threshold (Ct) values of each dilution were evaluated by qRT-PCR with each primer set to generate the standard curves.

Table 2 Primer sets used in this study and their efficiencies.

Gene	Accession number	Primer sequence (5′–3′)	Efficiency (%)	R2	Reference	
GAPDH	DQ404538.1	CACCTCAAGATCGTCAGCAA	100.97	0.9949	Chen et al. (2015)	
GCCGAAGTGGTCATGGAT	
B2M	DQ404542.1	GGTGGAGCAATCAGACCTGT	93.32	0.9984	Chen et al. (2015)	
GCGTTGGGAGTGAACTCAG	
IL-2	EU638316	CATGCCCAAGAAGGCTACAGAATTG	91.92	0.999	Sitt et al. (2008)	
GTGAATCTTGTTTCAGATCCCTTTAG	
IL-4	EU-638315	GGAGCTGCCTGTAGAAGACGTCTTTGC	99.25	0.9982	Sitt et al. (2008)	
CTTCATTCACAGAACAGGTCATGTTTGCC	
IL-10	AB775207	TGCTGGAGGACTTTAAGGGTTA	93.14	0.9986	Segawa et al. (2013)	
ATGAAGATGTCAAACTCACTCATG	
IL-12	EU638319	CAGACCAGAGCGATGAGGTCTTG	91.08	0.9999	Sitt et al. (2008)	
GGGCTCTTTCTGGTCCTTTAAGATA	
IFN-γ	EU638318	CAGAGCCAAATAGTCTCCTTCTACTTC	92.42	0.9976	Sitt et al. (2008)	
CTGGATCTGCAGATCATCTACCGGAATTTG	
TNF-α	EU638323	GAGGGAAGAGTTCCCAACTGGCTA	101.48	0.9934	Sitt et al. (2008)	
CTGAGTACTGAGGTTGGCTACAAC	

Extraction of RNA, synthesis of cDNA and qRT-PCR

Total RNA was extracted from blood samples by RNeasy® Mini Kit (Qiagen, Valencia, CA, USA) according to the manufacturer’s instructions. The RNA samples were treated with genomic DNA (gDNA) wipeout solution (Qiagen). Treated samples were then tested by qRT-PCR to confirm the absence of residue gDNA prior to cDNA synthesis. The QuantiTect® Reverse Transcription kit (Qiagen) was used for cDNA synthesis. The reverse transcription was conducted within 4 h after RNA extraction. The cDNA from each sample was stored at −20 °C for qRT-PCR. The qRT-PCR was performed on Mastercycler® ep realplex (Eppendorf, Hamburg, Germany). Each reaction contained 10 µl of SYBR® Advantage® qPCR Premix (Clontech, Mountain View, CA, USA), 7.2 µl of RNase/DNase-free sterile water, 0.4 µl of each 10 mM forward/reverse primers, and two µl of DNA template, and the final volume of each reaction was 20 µl. Two microliters of RNase/DNase-free sterile water was used as the non-template negative control. The thermocycle conditions were set as follows: initial denaturation at 95 °C for 30 s and 40 cycles of denaturation at 95 °C for 10 s, annealing at 60 °C for 20 s, and extension at 72 °C for 30 s with fluorescence detection. Furthermore, the melting curve analysis was performed at the end to identify non-specific amplification. All PCR protocols were performed in accordance with the Minimum Information for Publication of Quantitative Real-Time PCR Experiments (MIQE) guidelines (Bustin et al., 2009; Taylor & Mrkusich, 2014).

Time kinetics of mRNA expression levels of selected cytokines of cPBMCs

To evaluate the time kinetics of mRNA expression levels of selected cytokines in cPBMCs with ConA, the cytokine gene expression levels of cPBMCs with ConA (0.5 µg/ml) were determined by qRT-PCR ( N = 4). Freshly-isolated cPBMCs were seeded in 96-well plates at a density of 5 × 105 cells/well and incubated for 0, 4, 8, 12, 16, 20 and 24 h of culture in a humidified atmosphere of 5% CO2 at 37 °C. Then the cPBMCs were collected for subsequent mRNA extraction, complementary DNA (cDNA) synthesis, and qRT-PCR. The cPBMCs with 0 h incubation were used as control for the calculation of cytokine expression level by ΔΔCT method. In addition, the geometric means of two housekeeping genes (HKGs), glyceraldehyde 3-phosphate dehydrogenase (GAPDH) and β2-microglobulin (B2M), of each sample were determined and used to normalize the expression levels of target genes (Hellemans et al., 2007; Vandesompele et al., 2002).

Effects of C-AgNP20 on mRNA expression levels of selected cytokines of cPBMCs

The cPBMCs were seeded in 96-well plates at a density of 5 × 105 cells/well and exposed to sub-lethal doses of C-AgNP20 with 0.5 µg/ml ConA for 4 and 24 h of culture in a humidified atmosphere of 5% CO2 at 37 °C. After incubation, the cPBMCs were collected for subsequent mRNA extraction, cDNA synthesis, and qRT-PCR. PBMCs with 4 and 24 h incubation without C-AgNP20 treatment were used as control for the calculation of cytokine expression level by ΔΔCT method. In addition, the geometric means of two HKGs, GAPDH and B2M, of each sample were determined and used to normalize the expression levels of target genes (Hellemans et al., 2007; Vandesompele et al., 2002). The experiment was independently repeated twice in duplicate (N = 12).

Statistical analysis

In all experiments, the results from duplicates were averaged. To compensate for individual differences, the results at different concentrations of C-AgNP20 for each individual were calculated as percentages of the results of the control (exposed to 0 µg/ml C-AgNP20). In addition, Th1/Th2 ratios at different concentrations of C-AgNP20 were determined by the cytokine mRNA ratios of Th1 (IL-12 or INF-γ) and Th2 (IL-4) polarizing cytokines and then compared to the control. Our data were first checked by Shapiro–Wilk normality test and Brown-Forsythe test, and the results indicated that the assumptions of normality and/or equal variance were violated. Therefore, the Kruskal-Wallis Test (post hoc test: Dunn’s multiple comparison test) was subsequently performed on the data. A p value <0.05 was considered statistically significant, and the analysis was performed in Prism (GraphPad Software, La Jolla, CA, USA). All data were plotted on box-plot graphics. The bar in the middle of the box represented the second quartile (median), and the bottom and top of the box described the first and third quartiles. The whiskers showed that the 75th percentile plus 1.5 times IQR and 25th percentile minus 1.5 times IQR of all data, and any values that greater than these were defined as outliers and plotted as individual points. Asterisks above the boxplots indicated statistically significant differences compared to the control of each experiment.

Results

Characterization of C-AgNP20

The C-AgNP20 in complete RPMI-1640 media were spherical and close to 20 nm in diameter (Fig. 1). The size distributions and zeta potentials of C-AgNP20 (100 or 500 µg/ml) in 2 mM citrate buffer are illustrated in Table 1. The size distributions were 30.27 ± 0.18 (100%) and 29.64 ± 0.30 (100%) for C-AgNP20 at 100 and 500 µg/ml, respectively. The values of the zeta potential were −38.97 ± 1.33 and −44.2 ± 1.35 mV for C-AgNP20 at 100 and 500 µg/ml, respectively. Furthermore, the Poly-dispersity Indexes (PDIs) were 0.12 ± 0.00 and 0.11 ± 0.01, indicating that the composition of C-AgNP20 in the stock was in a single size mode without aggregates.

Figure 1 Characterization of C-AgNP20.

Representative TEM image of C-AgNP20 in complete RPMI-1640.

Sub-lethal dose of C-AgNP20 to the cPBMCs with or without ConA stimulation

The treatment of C-AgNP20 at 10 µg/ml significantly increased the ratios of late apoptotic/necrotic cells in cPBMCs with or without ConA stimulation. The ratios of early apoptotic and late apoptotic/necrotic cells of cPBMCs with different concentrations of C-AgNP20 as compared to the control are presented in Fig. 2. After 24 h of 10 µg/ml C-AgNP20 treatment, the ratios of late apoptotic/necrotic cells of cPBMCs significantly increased with (median ± interquartile range (IQR): 3.55 ± 3.42; p = 0.0073) or without ConA stimulation (median ± IQR: 1.78 ± 2.24; p = 0.0103). In contrast, no statistically significant increases in the ratios of apoptotic and late apoptotic/necrotic cells in cPBMCs were found after 24 h culture with 0.1 and 1.0 µg/ml C-AgNP20 treatments. Therefore, 0.1 and 1.0 µg/ml C-AgNP20 were defined as the sub-lethal doses for cPBMCs and used in the subsequent mRNA expression levels of selected cytokines in cPBMCs to C-AgNP20.

Figure 2 Cytotoxicity of C-AgNP20 on cPBMCs after 24 h of culture with or without ConA.

(A) Ratio of apoptotic cPBMCs in percentage between treatment and control without ConA. (B) Ratio of late apoptotic/necrotic cPBMCs in percentage between treatment and control without ConA. (C) Ratio of apoptotic cPBMCs in percentage between treatment and control with ConA. (D) Ratio of late apoptotic/necrotic cPBMCs in percentage between treatment and control with ConA. The bar in the middle of the box represents the median, and the bottom and top of the box describe the first and third quartiles. The whiskers show the 75th percentile plus 1.5 times IQR and 25th percentile minus 1.5 times IQR of all data, and any values that are greater than these are defined as outliers and plotted as individual points. Asterisks indicate statistically significant differences from the control (p < 0.05, Kruskal–Wallis Test).

qRT-PCR efficiency of each primer set

Amplification efficiency (E) values for selected HKGs and cytokine genes, including GAPDH, B2M, IL-2, IL-4, IL-10, and IL-12, IFN-γ, and TNF-α, ranged from 91.08 to 101.48% with R2 > 0.99. The results are summarized in Table 2.

Time kinetics of mRNA expression levels of selected cytokines of cPBMCs with ConA stimulation

The mRNA expression levels of IL-2 and TNF-α were significantly increased at 4 h of culture, gradually decreased from 8 to 20 h of culture, and then mildly but not significantly increased at 24 h of culture. The mRNA expression level of IFN- γ was significantly increased at 4 h of culture, gradually decreased at 8 and 12 h of culture, and then increased from 16 to 24 h of culture. In addition, IL-4, IL-10, and IL-12 were significantly increased at 4 h of culture and gradually decreased over time. Therefore, the time points chosen for the following experiments were 4 h and 24 h. All the results are illustrated in Fig. 3.

Figure 3 Time kinetics of mRNA expression levels of (A) IL-2, (B) IL-4, (C) IL-10, (D) IL-12, (E) IFN-γ and (F) TNF-α of cPBMCs with ConA.

The bar in the middle of the box represents the median, and the bottom and top of the box describe the first and third quartiles. The whiskers show the 75th percentile plus 1.5 times IQR and 25th percentile minus 1.5 times IQR of all data, and any values that are greater than these are defined as outliers and plotted as individual points. Asterisks indicate statistically significant differences from the control (p < 0.05, Kruskal–Wallis Test).

Figure 4 Effects of C-AgNP20 on mRNA expression levels of (A) IL-2, (B) IL-4, (C) IL-10, (D) IL-12, (E) IFN-γ and (F) TNF-α of cPBMCs with ConA.

The bar in the middle of the box represents the median, and the bottom and top of the box describe the first and third quartiles. The whiskers show the 75th percentile plus 1.5 times IQR and 25th percentile minus 1.5 times IQR of all data, and any values that are greater than these are defined as outliers and plotted as individual points. Asterisks indicate statistically significant differences from the control (p < 0.05, Kruskal–Wallis Test).

Figure 5 The ratios of Th1 and Th2 polarizing cytokines at 4 and 24 h of culture.

(A) Ratio of IFN-γ and IL-4; (B) ratio of IL-12 and IL-4. The bar in the middle of the box represents the median, and the bottom and top of the box describe the first and third quartiles. The whiskers show the 75th percentile plus 1.5 times IQR and 25th percentile minus 1.5 times IQR of all data, and any values that are greater than these are defined as outliers and plotted as individual points. Asterisks indicate statistically significant differences from the control (p < 0.05, Kruskal–Wallis Test).

Effects of C-AgNP20 on mRNA expression levels of selected cytokines in cPBMCs

At 4 h of culture, the mRNA expression level of IL-10 was significantly decreased (median ± IQR: 0.7864 ± 0.2355; p = 0.0049) at 1 µg/ml C-AgNP20, but no significant differences were observed in the mRNA expression levels of other cytokine genes at 0.1 or 1 µg/ml C-AgNP20 (Fig. 4). Following 24 h of culture with 1 µg/ml C-AgNP20, the mRNA expression levels of IL-2, IL-12, IFN-γ, and TNF-α were significantly decreased, but no significant difference was found in those of IL-4 and IL-10. Furthermore, the mRNA expression levels of IL-12 (median ± IQR: 0.8337 ± 0.2088; p = 0.0339) and IFN-γ (median ± IQR: 0.7894 ± 0.389; p = 0.0164) were also significantly decreased at 0.1 µg/ml C-AgNP20 (Fig. 4). The Th1/Th2 bias was defined by the ratios of Th1 and Th2 polarizing cytokines. The IFN-γ/IL-4 ratio was significantly decreased following 24 h of culture with 1 µg/ml C-AgNP20, and the IL-12/IL-4 ratio was significantly decreased following 4 or 24 h of culture with 0.1 or 1 µg/ml C-AgNP20 treatments. Overall, the in vitro cytokine responses of cPBMCs with C-AgNP20 treatments were biased toward Th2 cytokine response (Fig. 5).

Discussion

Our data indicated that the concentration of 10 µg/ml C-AgNP20 was lethal dose for cPBMCs after 24 h of culture. Although previous studies of human PBMCs have used a variety of AgNPs (including different sizes and coatings), the lethal dose of AgNPs to human PBMCs is generally higher than 10 µg/ml (Ghosh et al., 2012; Greulich et al., 2011; Huang et al., 2016a; Orta-Garcia et al., 2015; Paino & Zucolotto, 2015; Shin et al., 2007). Therefore, our data suggest that cPBMCs may be more vulnerable than human PBMCs to the cytotoxic effects of C-AgNP20. However, previous studies have demonstrated that the toxicity and physicochemical characteristics of AgNPs are associated with their surface coating and size (Kim & Ryu, 2013), and thus further investigation using the same AgNPs from the same manufacturer is necessary to compare the differences of susceptibility between cetaceans and humans. In addition, the negative effects of AgNPs with different sizes and coatings on the cPBMCs are also worth to be further studied.

It has been demonstrated that ConA (a selective T-cell mitogen) induces proliferative activity and gene expression of cytokines in bottlenose dolphins, but no information is available regarding the time course (Hofstetter et al., 2017; Segawa et al., 2013; Sitt et al., 2008). Previous studies on ConA-induced cytokine mRNA expression levels of cPBMCs only presented one or two time points (Segawa et al., 2013; Sitt et al., 2008). Sitt et al. (2008) quantified the ConA-induced cytokine mRNA expression levels of cPBMCs after 48 h of treatment, but the reason for choosing this time point was not explained. Their results showed that the mRNA expression levels of IL-2, IL-4, IL-12, and IFN-γ in cPBMCs are induced after 48 h of ConA stimulation, but those of IL-10 and TNF-α were not increased (Sitt et al., 2008). The other study demonstrated that the mRNA expression level of IL-10 in cPBMCs increased after 6 h of ConA stimulation (Segawa et al., 2013). Therefore, to apply appropriate time points for studying the effects of C-AgNP20 on the cytokine mRNA expression levels of cPBMCs, the time kinetics (from 0 to 24 h) of mRNA expression levels of IL-2, IL-4, IL-10, IL-12, IFN-γ and TNF- α in cPBMCs with ConA stimulation were investigated. Our data indicated that the mRNA expression levels of all cytokine genes were significantly increased at 4 h of ConA stimulation and then gradually decreased with time. A longer incubation time was not possible in our study because the cell numbers of isolated cPBMCs were insufficient.

Previous studies have investigated the negative effects of AgNPs on cytokines in human PBMCs (Franco-Molina et al., 2016; Greulich et al., 2011; Shin et al., 2007). Uncoated AgNPs (1.5 nm; 1 to 2.5 nm in diameter) significantly inhibited the phytohemagglutinin (PHA)-induced IL-5, IFN-γ, and TNF-α production respectively at concentrations ≥10μg/ml, ≥3μg/ml, and ≥3μg/ml in human PBMCs (Shin et al., 2007). It was reported that uncoated AgNPs (100 nm; 90 to 190 nm in diameter) at 0.0175 µg/ml can inhibit both PHA and ConA-induced IL-2 production in human PBMCs (Franco-Molina et al., 2016). Furthermore, polyvinylpyrrolidone (PVP)-coated AgNPs (75 ± 20 nm) of five to 20 µg/ml significantly increased the generations of IL-6 and IL-8 but significantly decreased the release of IL-1ra from human PBMCs, while PVP-coated AgNPs did not affect the productions of IL-2, IL-4 and TNF-α (Greulich et al., 2011). As mentioned above, the effects of AgNPs on cytokine production in human PBMCs remain inconclusive.

The mRNA expression levels of IL-4 and IFN-γ were mildly increased and that of IL-12 was seemingly unaffected at 4 h of C-AgNP20 treatment. IL-4, as a polarizing Th2 cytokine, is mainly produced by T cells (especially the Th2 subset) and mast cells, and it promotes the differentiation of naïve T cells to Th2 cells, stimulates the growth and differentiation of B cells, and induces class switching to IgE, which may promote allergic responses (Owen et al., 2013; Tizard, 2013b). IFN-γ, as a polarizing Th1 cytokine and a key mediator of cell-mediated immune response, is produced by Th1 cells, cytotoxic T cells, and NK cells. The major functions of IFN-γ are enhancement of Th1 differentiation, inhibition of Th2 differentiation, and activations of NK cells and macrophages (Owen et al., 2013; Tizard, 2013b). IL-12 is also a polarizing Th1 cytokine and is produced by dendritic cells, monocytes, macrophages and B cells. IL-12 induces differentiation of Th1 cells, increases IFN-γ production by T cells and NK cells, and enhances NK and cytotoxic T cell activity (Owen et al., 2013; Tizard, 2013b). This mixed pattern of Th1 and Th2 cytokines may be indicative of a mixed Th1/Th2 cytokine response of cPBMCs at 4 h of C-AgNP20 treatment. However, considering the significant decrease in the IL-12/IL-4 ratio, Th2 cytokine response is still predominant in cPBMCs following 4 h of C-AgNP20 treatment. The mRNA expression levels of IL-12 and IFN-γ were significantly decreased by 0.1 or one µg/ml C-AgNP20, and that of IL-4 was seemingly unaffected, in cPBMCs following 24 h of culture. The significantly decreased Th1/Th2 (i.e., IFN-γ/IL-4 and IL-12/IL-4) ratios suggested that the immune response of cPBMCs following 24 h of C-AgNP20 treatment is Th2 biased.

Furthermore, the mRNA expression level of TNF-α was significantly decreased by 1 µg/ml C-AgNP20 after 24 h of culture. TNF-α is a cytokine specifically useful to measure the inflammatory state of an animal and it is primarily produced by macrophages and both Th1 and Th2 cells in response to both acute and chronic conditions (Eberle et al., 2018). Previous studies have demonstrated that TNF-α is a contributing factor in the inflammatory response against infection of intracellular micropathogens such as Plasmodium spp., T. gondii, Leishmania major, and Trypanosoma spp. (Korner et al., 2010). Hence, our data indicate that C-AgNP20 induced a Th2 biased immune response and suppressed the mRNA expression level of TNF-α in cPBMCs, which may weaken the cellular immune response and further impair the immunity against intracellular organisms and virus. Similar Th2 immune response was observed in other studies that evaluated the expression of cytokines in different cetacean tissues (Jaber et al., 2010). A variety of infections caused by intracellular pathogens in cetaceans have been reported and may be associated with the mass stranding events of cetaceans (Cvetnic et al., 2016; Domingo et al., 1990; Domingo et al., 1992; Dubey et al., 2007; Dubey et al., 2008; Mazzariol et al., 2016; Mazzariol et al., 2017). In addition, previous studies suggested that Ag contamination exists in all aspects of the marine ecosystem, and cetaceans may have been negatively affected by Ag contamination (Becker et al., 1995; Caceres-Saez et al., 2013; Chen et al., 2017; Dehn et al., 2006; Kunito et al., 2004; Li et al., 2018a; Mendez-Fernandez et al., 2014; Reed et al., 2015; Rosa et al., 2008; Seixas et al., 2009; Woshner et al., 2001). The direct correlation between the infection of intracellular pathogens and the severity of Ag contamination in cetaceans is worth studying.

Following 4 h of 1 µg/ml C-AgNP20 treatment, the mRNA expression level of IL-10 was significantly decreased and that of IL-2 was mildly increased. In other words, mRNA expression levels of IL-2 and IL-10 were respectively upregulated and downregulated by C-AgNP20 in cPBMCs. Subsequently, the mRNA expression level of IL-2 was significantly decreased, and that of IL-10 seemingly unaffected, in cPBMCs following 24 h of treatment of 1 µg/ml C-AgNP20. IL-2, which is produced by activated T cells, can stimulate proliferation and differentiation of T and B cells and activates NK cells (Owen et al., 2013; Tizard, 2013b). However, a growing body of evidence has indicated that IL-2 is crucial for the development and function of regulatory T cells (Treg cells), which secrete effector cytokines, such as IL-10, to control and modulate the immunity to self, neoplasia, microorganisms, and grafts (Owen et al., 2013; Pérol & Piaggio, 2016). Considering the roles of IL-2 and IL-10 in immune tolerance, it is speculated that C-AgNP20 may play a significant role in peripheral immune tolerance by regulating the balance between IL-2 and IL-10 (Pérol & Piaggio, 2016; Veiopoulou et al., 2004).

The effect of C-AgNP20 on the ConA-induced mRNA expression levels of the selected cytokines in cPBMCs is mainly inhibitory. A previous study found that PVP-AgNPs (10, 25, 40, 45, and 110 nm in diameter) could bind to RNA polymerase, disturb the process of RNA transcription, and thus decreased the overall RNA synthesis in mouse erythroid progenitor cells (Wang et al., 2013). Although the down-regulation of mRNA expression levels may be associated with decreased RNA synthesis due to the direct interaction between C-AgNP20 and RNA polymerase, it cannot fully explain the unaffected Th2 cytokines (IL-4 and IL-10) of cPBMCs in this study. On the other hand, the ConA-induced proliferative activity of cPBMCs is inhibited by 0.1 and 1.0 µg/ml C-AgNP20 (Li et al., 2018b), and this phenomenon may be associated with the decreased mRNA expression levels of IL-2, IL-12, IFN-γ, and TNF-α and/or a suppressive effect on DNA/RNA synthesis induced by ConA. Further investigation on the underlying mechanism of AgNPs in cetacean leukocytes is important to ascertain the negative health impact caused by AgNPs on cetaceans, and such investigation would improve the understanding of the potential hazards of AgNPs to environmental condition and human health.

Furthermore, although the biodistribution of AgNPs or Ag in cetaceans is still undetermined, previous in vivo studies of AgNPs by oral exposure in laboratory rats demonstrated that the Ag concentration in the liver is approximately 10 times higher than that in the blood or plasma (Lee et al., 2013; Loeschner et al., 2011; Van der Zande et al., 2012). Based on these animal models, it is presumed that the Ag concentrations in the blood of cetaceans may range from 0.01 to 72.6 µg/ml (Chen et al., 2017; Li et al., 2018a). Although previous studies have indicated that the status of AgNPs in the aquatic environment is complicated and variable (i.e., the concentrations of AgNPs and other Ag/Ag compounds are still undetermined in cetaceans)(Levard et al., 2012; Massarsky, Trudeau & Moon, 2014), our data suggest that cetaceans may be negatively affected by AgNPs.

Conclusions

The present study has demonstrated: (1) the sublethal dose of C-AgNP20 to cPBMCs (≤1 µg/ml), (2) the time kinetics of mRNA expression levels of selected cytokines in cPBMCs, and (3) the inhibitory effect of C-AgNP20 (0.1 and 1 µg/ml) on the mRNA expression levels of selected cytokines of cPBMCs with evidence of Th2 cytokine bias. Taken together, C-AgNP20 may suppress the cellular immune response and thus inhibit the immunity against intracellular microorganisms in cetaceans.

Supplemental Information

Supplemental Information 1 The size distribution and zeta pontential of the 100 and 500 µg/ml C-AgNP20 in 2 mM citrate buffer (pH 7.4)

Click here for additional data file.

Supplemental Information 2 Cytotoxicity of of C-AgNP20 in cPBMCs with or without ConA stimulation

Click here for additional data file.

Supplemental Information 3 qPCR efficiencies of each primer set (ct values)

Click here for additional data file.

Figure S1 qPCR efficiencies of each primer set

Click here for additional data file.

Supplemental Information 4 Time kinetics of mRNA expression levels of selected cytokines of cPBMCs

The cPBMCs with 0 h incubation were used as control for the calculation of cytokine expression level by ΔΔCT method. In addition, the geometric means of two housekeeping genes (HKGs), glyceraldehyde 3-phosphate dehydrogenase (GAPDH) and β2-microglobulin (B2M), of each samples were determined and used to normalize the the expression levels of target genes. The numbers in the sheet were the expression levels.

Click here for additional data file.

Supplemental Information 5 Effects of C-AgNP20 on mRNA expression levels of selected cytokines of cPBMCs (4 h)

The cPBMCs with 4 h incubation without C-AgNP20 treatment were used as control for the calculation of cytokine expression level by ΔΔCT method. In addition, the geometric means of two HKGs, GAPDH and B2M, of each samples were determined and used to normalize the the expression levels of target genes

Click here for additional data file.

Supplemental Information 6 Effects of C-AgNP20 on mRNA expression levels of selected cytokines of cPBMCs (24 h)

The cPBMCs with 24 h incubation without C-AgNP20 treatment were used as control for the calculation of cytokine expression level by ΔΔCT method. In addition, the geometric means of two HKGs, GAPDH and B2M, of each samples were determined and used to normalize the the expression levels of target genes.

Click here for additional data file.

We thank the personnel of Farglory Ocean Park for blood sample collection and storage, the dolphins in Farglory Ocean Park for donating their blood, and Dr. Bang-Yeh Liou for blood sample transportation.

Additional Information and Declarations

Competing Interests

Author Contributions

Animal Ethics

Data Availability

Chieh Lo is the veterinarian of Farglory Ocean Park, Hualien, Taiwan.

Wen-Ta Li conceived and designed the experiments, performed the experiments, analyzed the data, prepared figures and/or tables, authored or reviewed drafts of the paper, approved the final draft.

Lei-Ya Wang performed the experiments, analyzed the data, prepared figures and/or tables, authored or reviewed drafts of the paper, approved the final draft.

Hui-Wen Chang, Wei-Cheng Yang, Victor Fei Pang and Meng-Hsien Chen conceived and designed the experiments, contributed reagents/materials/analysis tools, authored or reviewed drafts of the paper, approved the final draft.

Chieh Lo performed the experiments, authored or reviewed drafts of the paper, approved the final draft, collected the blood samples from captive dolphins.

Chian-Ren Jeng conceived and designed the experiments, analyzed the data, contributed reagents/materials/analysis tools, authored or reviewed drafts of the paper, approved the final draft.

The following information was supplied relating to ethical approvals (i.e., approving body and any reference numbers):

All procedures involving animals were conducted in accordance with international guidelines, and the protocol had been reviewed and approved by the Council of Agriculture of Taiwan (Approval number 1051700175).

The following information was supplied regarding data availability:

The raw data are provided in the Supplemental Files.

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
