# Peer review of "Th2 cytokine bias induced by silver nanoparticles in peripheral blood mononuclear cells of common bottlenose dolphins (Tursiops truncatus)"

_PeerJ, doi:10.7717/peerj.5432_

## Round 0.1 · original submission · Minor Revisions

Based on the opinions of three reviewers, this manuscript is being returned for minor revisions. Overall reviewers agreed that the manuscript was well written, that the work was thorough and well done, and that appropriate conclusions have been made. Note, although I appreciate reviewer #2's concerns about the reference genes, because the authors used two reference genes and took the geometric mean as is the current standard practice, I don't believe the authors need to rerun their qPCR. That being said, please address all reviewer comments carefully and thoroughly.

·

Basic reporting

The paper is well written and even though is not totally ground-breaking, it is an important contribution to the nanotoxicity field. It provides new insight on how the nanomaterials can affect different organisms, and those differences can be seen even at cellular level.
The conclusions are well supported by the results.

Experimental design

The methods used are appropiate for the study.

Validity of the findings

Again, the conclusions are well supported by the data presented in the results section. No invalid extrapolations or specutalions were done (as tempting as it can be).

Additional comments

The paper is well written and even though is not totally ground-breaking, it is an important contribution to the nanotoxicity field. It provides new insight on how the nanomaterials can affect different organisms, and those differences can be seen even at cellular level.

The discussion can be improved. The information presented is all relevant, but sometimes it seems repetitive, making this section unnecessarily long. In general, try to be more concise, to the point.

I would also add some discussion on the effects of nanomaterials on the immune system of other marine organisms. For example, it has been demonstrated that nanomaterials can negatively affect the function of hemocytes in mussels (Canesi et al, 2012 Mar.Env. Res., 76:16, and related publications). Eventhough the organisms are very different and their immune systems have different modes of action, it would be illustrative to discuss how nanomaterials affect the immune system, making aquatic orgnisms more susceptible to infections.

Specific questions, suggestions and corrections are described below:

Line 130 - Please define ConA and briefly introduce the concept.

Line 158 - ug/ml, change the micro symbol.

Line 312-313 10 ug/l is not higher than 0.1 ug/l; please rephrase what you mean. It is well-known that the toxicity of nanomaterials depends on multiple factors, such as the properties of the nanomaterials (including the chemical composition) and the conditions of the experiment. It is briefly mentioned that the studies cited used a "variety of AgNPs". Though the specifics are discussed later in the paper, please briefly mention what was different: was it the composition? was it size? Also, briefly discuss if the experimental conditions were different and how that might influence the results.

Line 316-323 Please be more concise. For example: "It has been demonstrated that ConA induces proliferative activity and gene expression of cytokines in bottlenose dolphins, but no information is available regarding the time course."

Line 332 - What do you mean by "escalating trend"? In Figure 2 I do not see significant differences between time 4, 16 and 24.

Line 395 - Change "inhibition" to "inhibitory".

Line 416-417 - Change "have been" to "be". The likelyhood of exposure to AgNPs might increase in the coming years as its use in commercial products keeps increasing.

Reviewer 2 ·

Basic reporting

1. Clear English language is used throughout with a few minor exceptions.
There are some issues of redundancy within sentences throughout the paper.
Line 394-395: major… mainly… use of both of these terms to describe the effect of C-AgNP20 in the same sentence is redundant
Line 416: “may” used twice in sentence
In line 111, “Homology” rather than “coverage” would be more appropriate here.
In line 332, authors used the word “escalating” which means increasing. However, the cytokines are decreasing over the time periods mentioned.
2. It is not necessary to present the abbreviation for silver nanoparticles or peripheral blood mononuclear cells in the article title.
3. There are some instances of non-English characters used in the supplemental materials. If possible, these should be changed to English to correspond to the text of the manuscript.
4. Indent line 381.
5. Intro and background sufficiently show context and literature comprehension.
6. Structure follows PeerJ standards.
7. Figures are clear and well-described.
8. All raw data has been provided.

Experimental design

1. Original research falls within the scope of the journal.
2. Important research question addresses impact of silver nanoparticles on cPBMCs using an ex vivo system previously defined by other research groups.
3. While there have been several studies examining the effect of C-AgNP20 on human blood cells, Poirier et al. 2016 actually noted differences between AgNP20 and AgNP70 as regards their biological properties in modulating functions of human neutrophils. As it is not clear how different sized NPs may impact biological function of dolphin cells, this should be examined at some point.
4. What was the free Ag concentration of the AgNPs (see for example Kwok et al. 2012. Aquatic Toxicology 120-121:59-66)
5. There is a primary concern regarding the use of GAPDH as a housekeeping gene. The authors have referenced Chen et al (2015) that found that GAPDH was a fairly poor housekeeping gene, ranking 10th among 15 tested. It is also likely to vary with cells’ metabolic state, which is likely to be affected by mitogenic stimulation. How were your housekeeping genes selected and validated?

Validity of the findings

1. Statistics were performed with appropriate rigor.
2. Conclusions are clear and do not, in general, overstate the findings.

Additional comments

1. Line 44: Add the word “ratio” after “IFN-γ/IL-4” or state “Ratio of IFN-γ/IL-4”
2. Line 187: “(n=8)” seems out of place here.
3. Line 312-313: the range of doses given is from 0.1 μg/mL to 50 μg/mL, but the sentence says the lethal dose is generally higher than 10 μg/mL. This implies that some studies have shown lethality at lower doses. Then how can the conclusion be made in lines 314-315.
4. Line 320-321 state that previous studies have only presented a single timepoint. One of the papers you cited, Hofstetter et al., presented data from both 24 h and 48 h.

Reviewer 3 ·

Basic reporting

The paper looks clear and unambiguous, and the references are updated

Experimental design

The paper submitted can fit with the scope of the journal

Validity of the findings

This paper shows novelty that could be used to get information about silver nanoparticles presence in active stranding cetaceans. Additionally, conclusions are well stated and limited to supporting results.

Additional comments

This paper evaluates the cytokine responses of cPBMCs to citrate silver nanoparticles by quantitative RT-PCR. This is a nice study and give new elements in the knowledge of the posible impact of AgNPs in marine mammals. However, there are some considerations before acceptance.
General Comments
Some sections of the manuscript are not well written due to unclear sentence structure or missing words. Therefore, introduction section is too long and there are some explanations that could be avoided. Important care has been taken in describing the methodologies accurately. Conclusions and perspectives are well stated, linked to original research question and limited to supporting results. Nonetheless, there are some speculations in the discussion section that must be better explained or deleted. Figures show excellent information that is very helpful to the readers.
Some specific comments have been included below to aid the authors and explain the statements above.
In introduction section,
Line 67-71: Please shorten and link the two sentences.
Line 75: Please add “Nevertheless, to date the potencial toxicity…”
Line 87-95: Please redone to avoid some repetitions.
Line 130-134: Please redone and move the entire paragraph to discussion section
Line 137: This paragraph must be redone and last sentence moved to material and methods section

Materials and Methods,
Line 161-170: please shorten. Do not repeat the protocol twice
In results and discussion section,
Line 299: in relation to diminish mRNA expression levels of cytokines at different time (4 h vs 24 h culture). It would be very useful to know the precise time between the interval (4-24H) when it happens.
Line 314: Please explain better this sentence since you do not give enough information to demonstrate that cPBMC in dolphins are weaker than human PBMCs to the toxic effect of c-AgNP.
Line 366-368: Please change “TNF-alpha, a central inducer …(Owen et al., 2013) by “ TNF-alpha is a cytokine specifically useful to measure the inflammatory state of an animal and it is primarily produced by macrophages and both Th1 and Th2 cells in response to both acute and chronic conditions (Eberle et al. 2018. Development and testing of species- specific ELISA to measure IFN-gamma and TNF-alpha in botlenose dolphins. PLOS one , volume 13, issue 1).
Line 373: after “organism and virus.” please add “Similar Th2 immune response was observed in other studies that evaluated the expression of cytokines in different cetacean tissues (Jaber et al. 2010. Cross-reactivity of Anti-human …Journal of Comparative Pathology, volume 143, 45-51).
Line 375: Please “Dubay et al. 2007, after Cvetnic et al 2016). In addition, you must include important mass stranding by Morbillivirus infection (Domingo et al. 1990. Morbillivirus in dolphins. Nature, 348, pp21; Domingo et al. 1992. Pathologic and immunocytochemical studies of morbillivirus infection in striped dolphins. Veterinary Pathology, 29: 1-10).
Line 376-379: please add some references that show other locations, no just in the Pacific Ocean.

---

## Round 0.2 · accepted · Accept

The authors have sufficiently addressed reviewers' concerns and have revised their manuscript accordingly.

#